# Cell-morphodynamic phenotype classification with application to cancer metastasis using cell magnetorotation and machine-learning

Remy Elbez[1☯], Jeff Folz[2☯], Alan McLean[3], Hernan Roca[4], Joseph M. Labuz[5], Kenneth J. Pienta[6], Shuichi Takayama[7], Raoul Kopelman[1,2,3]*

1 Applied Physics Program, University of Michigan, Ann Arbor, Michigan, United States of America, 2 Biophysics Program, University of Michigan, Ann Arbor, Michigan, United States of America, 3 Department of Chemistry, University of Michigan, Ann Arbor, Michigan, United States of America, 4 Department of Urology, University of Michigan School of Medicine, Ann Arbor, Michigan, United States of America, 5 Department of Biomedical Engineering, University of Michigan, Ann Arbor, Michigan, United States of America, 6 Department of Urology, The James Buchanan Brady Urological Institute, Johns Hopkins Hospital, Baltimore, Maryland, United States of America, 7 Department of Biomedical Engineering, Georgia Institute of Technology, Atlanta, Georgia, United States of America

☯ These authors contributed equally to this work.
* kopelman@umich.edu

**Data Availability Statement:** Data and Python scripts used for the analysis can be found at the

## Abstract

We define cell morphodynamics as the cell's time dependent morphology. It could be called the cell's *shape shifting ability*. To measure it we use a *biomarker free*, dynamic histology method, which is based on multiplexed *Cell Magneto-Rotation* and *Machine Learning*. We note that standard studies looking at cells immobilized on microscope slides cannot reveal their shape shifting, no more than pinned butterfly collections can reveal their flight patterns. Using cell magnetorotation, with the aid of cell embedded magnetic nanoparticles, our method allows each cell to move freely in 3 dimensions, with a rapid following of cell deformations in all 3-dimensions, so as to identify and classify a cell by its *dynamic morphology*. Using object recognition and machine learning algorithms, we continuously measure the real-time *shape dynamics* of each cell, where from we successfully resolve the inherent broad heterogeneity of the *morphological phenotypes* found in a given cancer cell population. In three illustrative experiments we have achieved clustering, differentiation, and identification of cells from (A) two distinct cell lines, (B) cells having gone through the *epithelial-to-mesenchymal transition*, and (C) cells differing only by their *motility*. This microfluidic method may enable a *fast screening* and identification of *invasive cells, e.g., metastatic cancer cells*, even in the absence of biomarkers, thus providing a rapid diagnostics and assessment protocol for effective personalized cancer therapy.

## Introduction

Despite much progress over the last century, cancer remains one of the leading causes of death globally [1]. Its lethality is overwhelmingly due to metastasis, the process by which cells from

University of Michigan's DeepBlue data library
https://doi.org/10.7302/513f-1h23.

**Funding:** The authors wish to thank the National Institute of Health/National Cancer Institute IMAT program for financial support through an NIH-NCI (IMAT) grant R21 CA160157 (RK), as well as NIH grants CA136829 (ST), R01CA186769 (RK) and 1R01CA250499 (RK). JML gratefully acknowledges support from the University of Michigan Tissue Engineering and Regenerative Medicine Training Program (NIH T32-DE007057), a US Department of Education GAANN fellowship, and the University of Michigan Microfluidics in Biomedical Sciences Training Program (NIH T32 EB005582-05). The funders had no role in study design, data collection and analysis, decision to publish, or preparation of the manuscript.

**Competing interests:** The authors have declared that no competing interests exist.

the original cancerous tumor leave their micro-environment (TME) and disseminate to colonize new tissues [2]. During this *metastatic process*, separated single cells, or multi-cellular clusters, migrate through the *extra-cellular matrix* (ECM) surrounding the tumor, passing through the endothelium into the bloodstream [3, 4]. Upon entering the bloodstream, cells and clusters are buffeted by hemodynamic forces on the range of 4–30 dynes/cm$^2$ [5–7]. Additionally, these cells must contend with immunological insults and collisions with red blood cells. Having survived under these conditions, cancer cells must latch onto epithelial cells and extravasate into "foreign" tissue, so as to seed a secondary tumor.

To complete the aforementioned challenges, *metastatic cells* must express entirely *different phenotypes* than their stationary counterparts. Specifically, the *epithelial to mesenchymal transition* (EMT) permits the relatively stationary, epithelial cells of solid tumors to obtain the mobility required to *intravasate* and exit the primary tumor, and eventually intravasate at a new tissue location. The EMT may be induced without any gene mutations [8]. It has been observed that the post-EMT *amoeboid*-like cells can significantly increase the *metastatic potential* of the tumor [9]. It has also been reported that *morphological changes* can be used to identify cells having undergone the EMT [10, 11]. Morphology has been linked to cell cycle progression, cell-matrix adhesion properties, gene expression patterns, aging, chemo-sensitivity, and chemo-resistance [12, 13]. Morphology has also been used to predict the *metastatic potential* of both osteosarcoma [14, 15], and breast cancer [16]. Thus, morphology *with its dynamics* presents an attractive option for evaluating cancer progression. Here we emphasize the dynamic aspects, i.e., the *cancer cell's morphodynamics*, that is its time-dependent morphology (or shape shifting ability).

To date, most studies of cell morphology have focused on plated, adherent cell lines. Even though clear morphological distinctions can be discerned among cells when plated, the mere two-dimensional plating process might change their phenotypes and thus alter the quality of the diagnostics [17–22]. Furthermore, tumor grading by professional histologists is characterized by poor reproducibility and accuracy [23]. The goal of these studies has been to correlate genetic features with morphological ones. Wu et al. demonstrated that isolated morphological sub-phenotypes were predictive of tumorigenic and metastatic potentials [24]. In a separate study, the same group showed that metastatic cells possessed more homogenous heritable morphological traits [25]. Another study demonstrated that oncogenesis and metastasis were associated with characteristic changes in morphology [11, 26]. While studies have been conducted that differentiate cancerous from non-cancerous cells [27, 28], we have extended this analysis to compare metastatic with non-metastatic cell types. It would be of significant utility if a morphodynamic classification system could be built for suspended cells, such as circulating tumor cells (CTCs) or those harvested from a biopsy. To realize this goal, we use magnetic nanoparticles so as to trap, suspend, and rotate cells that are captured in a microfluidic chamber [29]. Magnetorotation prevents cells from adhering the microwells and permits exploration of their morphodynamic space.

To examine the cellular morphodynamics, we combine *cell magneto-rotation* with *machine learning* and we show that this approach may allow one to probe both cell motility as well as morphological expression. Machine learning has lent itself to many medical applications and removes the subjectivity of a histologist's analysis [24, 30]. In our approach, green fluorescent protein (GFP) expressing cancer cells are activated by endosomic uptake of magnetic nanoparticles, and are then loaded into a microfluidic device that contains an array of microwells where they remain non-adherent while rotating in an oscillating magnetic field [31, 32]. This enables 3-dimensional cell deformations in which the cells explore and express their morphological phenotype. Most of the device's microwells contain just one single cell; and in each such microwell the single cell is free to take *any* of the shapes that exist in its morphological

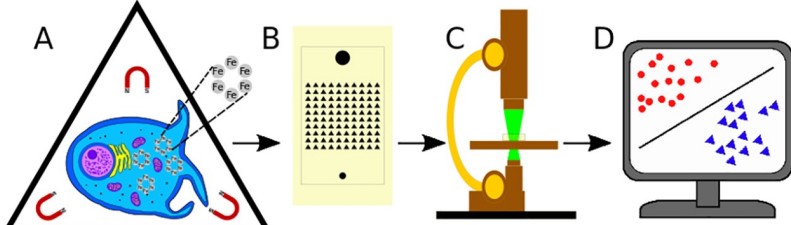

**Fig 1. A schematic summary of the cell morphodynamics protocol.** A: A captured cell expresses its morphodynamic phenotype while being gently rotated in a magnetic field, enabled by endocytosis of magnetic nanoparticles. B: The microfluidic device contains an array of triangular microwells designed so as to capture individual cells, in spaces large enough for cells to rotate freely. C: Rotating cells are fluorescently imaged on an environmentally controlled microscope stage. D: Cell images are converted by CellProfiler into parameters, used by Machine Learning algorithms to provide cellular clustering, classification, and analysis.

space. After taking fluorescent images of these cells, we combine *object recognition* and *machine learning* algorithms, so as to first *reject* information from multi-occupancy or empty microwells, and then to *differentiate, cluster, and identify* each of the cells by its *morphological profile* (Fig 1). We found that cells having undergone the EMT could be distinguished from control cells, which demonstrates the *morphodynamic equivalent of a change in protein expression*. Furthermore, highly migratory cells were found to be morphodynamically distinct from a control population. This new *machine learning* (ML) based method appears to have the potential to map and classify the *morphodynamic distribution* of a given cell population, and thus to provide information on the degree of *morphological plasticity* of a tumor cell's population. The latter may be related to the tumor's lethality. We have thus used this method here to demonstrate the strong relationship between a cell's morphological and biological behaviors.

## Results

To test our technique, we started with an example system: distinguishing between two well characterized breast cancer cell lines of unequal metastatic potential (MDA-MB-231 and MCF-7), using the *AdaBoost* algorithm [33]. The algorithm is capable of correctly identifying cells with a probability of 99%, both in terms of true positives and true negatives, for both phenotypes, as shown in Fig 2A. This means that, on average, our method has a false positive rate below 0.01, and a false negative rate below 0.01 as well (Fig 2A). Using *principal component analysis*, we can project the morphological measurements of the cells onto those principal components that explain most of the data's variance (Fig 2B). Doing so shows that distinct cell phenotypes are clustered together in a manner that allows for separation and classification. Naturally, this does not exclude partial overlap between distributions.

In order to test the capacity of our method, we also reduced the amount of training examples of the target cells (MDA-MB-231, being the most aggressive cells) in the general population (being represented by the MCF-7 cells). Surprisingly, even at a ratio of 1/1500 the algorithm is still capable of learning to differentiate the two cell lines, with an f1 score of 0.965. As scores for the reduced number of examples from one of the two categories remain high, it is instructive to look at how the *standard deviation* of the score changes (Fig 2C). We see that reducing the presence of one target population largely contributes to results with a higher variance; in other words, outliers, even small ones, have a larger effect for smaller sample sizes. One of the reasons for this is that having fewer examples for one class contributes to overfitting that particular class, and thus brings a poorer discrimination power when more diverse examples of the same class are presented to the computer.

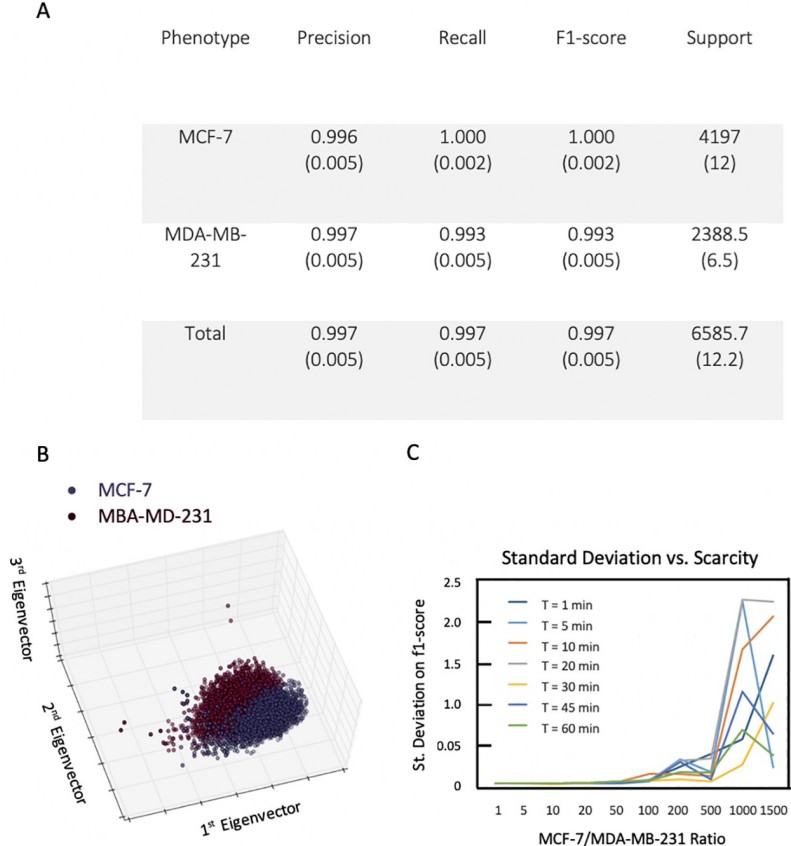

| Phenotype | Precision | Recall | F1-score | Support |
|---|---|---|---|---|
| MCF-7 | 0.996<br>(0.005) | 1.000<br>(0.002) | 1.000<br>(0.002) | 4197<br>(12) |
| MDA-MB-231 | 0.997<br>(0.005) | 0.993<br>(0.005) | 0.993<br>(0.005) | 2388.5<br>(6.5) |
| Total | 0.997<br>(0.005) | 0.997<br>(0.005) | 0.997<br>(0.005) | 6585.7<br>(12.2) |

**Fig 2. A 'proof of concept' classification task by morphodynamics with two distinct cell lines: MCF-7 (low metastatic potential, epithelial) and MDA-MB-231 (high metastatic potential, mesenchymal).** A: The detection power for MCF-7 vs. MDA-MB-231 after 1 minute (one image is captured every minute) using Adaboost. We observe high precision and recall, in both classes, indicating that the algorithm is robust for the two phenotypes, that the rate of false positives is low, and that we are capturing nearly every cell in each group (support designates the number of cells in a given group). B: Projections of the measured data onto the first three eigenvectors (principal components) reveals distinguishable clusters for the two cell lines: MCF-7 (blue) and MDA-MB-231 (red). Each point in the eigenvector space represents a single cell. C: A plot of the f1-score's standard deviation as a function of the MCF-7/MDA-MB-231 ratio. Colored lines indicate how long the cells were imaged. By increasing the number of scans, the classifier can become more confident in the phenotype of a particular cell, allowing us to distinguish the artificially rare subpopulation of MDA-MB-231 cells, even as their relative abundance becomes less than 0.1% of the population.

Having distinguished cells from two distinct lineages, we next utilized our method to analyze cells that originally shared the same phenotype, but had undergone a significant change, i.e., the *epithelial to mesenchymal transition* (EMT). To do so, we forced a human prostate cancer cell line, PC-3, to go through the EMT, yielding the cell line HR-14 [34]. As can be seen in Fig 3A, when using the representation of the cell measurements in the eigenvector space after only one imaging sequence, the two populations of cells are readily separable. To determine if any *sub-phenotypes* existed among these populations, we ran unsupervised *k-means clustering*, in which we partition our observations into k clusters, with each observation falling into the cluster with the nearest mean [35]. We chose k so as to maximize cluster homogeneity, with a strict constraint that all clusters must have a homogeneity greater than 0.95. In doing so, we found seven (7) sub-phenotypes (Fig 3B). Therefore, with our microfluidics imaging method, just using Cell Magneto-Rotation (CMR) coupled with unsupervised clustering techniques, we could identify and discriminate cells that went through an EMT. Furthermore, we were able to

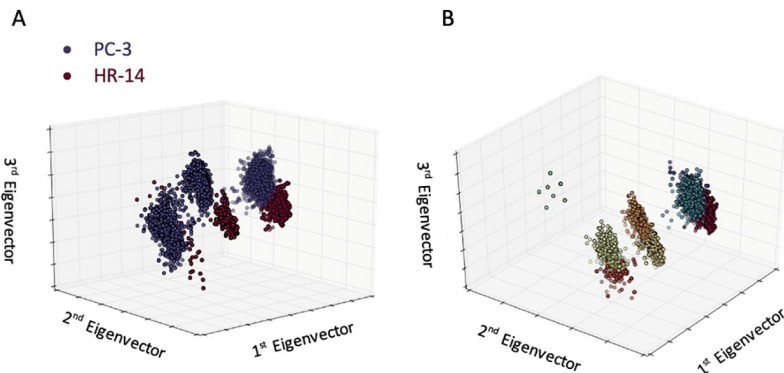

**Fig 3. Unsupervised analysis of PC-3 (epithelial) and HR-14 (mesenchymal) cell lines using *k-means clustering*.** A: The projection of PC-3 (blue) and HR-14 (red) cells onto the first three eigenvectors. Notably, when training the computer, it gets the information about each cell being either PC-13 or HR-14. Before any machine learning has been done, we can see that the two cell lines clearly cluster into distinguishable groups. Interestingly, we find that the clusters are not continuous and the emergence of *morphological sub-phenotypes* is apparent. B: The results of *k-means clustering* with the constraint that each cluster must maintain a homogeneity score greater than 0.95. We find that the constraint for highly pure clusters results in the identification of 7 sub-phenotypes.

identify the presence of *sub-phenotypes*, though their physiological manifestation remains uncharacterized.

Finally, we sought to detect subpopulations of cells within a given population. To do so, we separated those MDA-MB-231 cells that had a *higher motility* than the rest of the population, using a *Boyden chamber* [36]. We compared the data collected from these highly motile cells with those from the cells that had failed to migrate (Fig 4). Our results show that, using just a *morphodynamic analysis* with k-means clustering, we were able to easily distinguish the highly motile subpopulation of MDA-MB-231 cells from that of the general population. Our homogeneity score stands at 0.96 for 7 clusters, which means that the clusters have a very high purity. As a consequence, each cluster is composed, almost exclusively, of cells that have the same

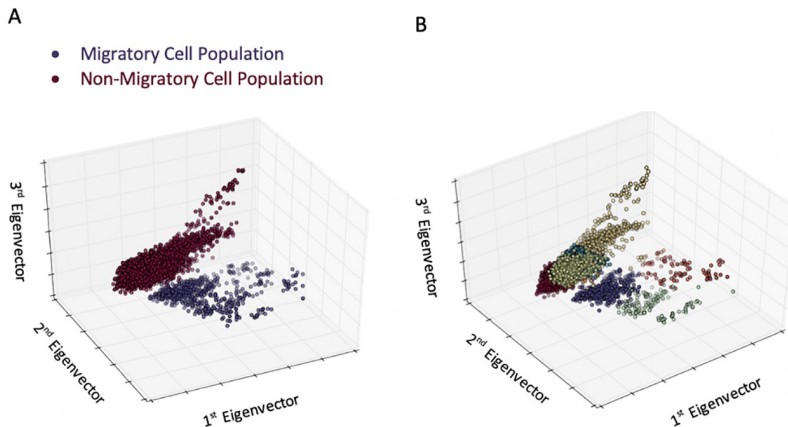

**Fig 4. Unsupervised analysis of highly motile cells separated from the bulk population chamber via a *Boyden chamber*.** A: Projections of migratory (blue) and non-migratory (red) MDA-MB-231 cells. The migratory and non-migratory fractions segregate into distinct clusters of cells, with many individual or small clusters of cells expanding into the periphery, away from the main clusters, indicating the presence of morphological sub-phenotypes. B: Using the *k-means clustering* algorithm with the strict criterion that clusters must have a homogeneity greater than 0.95, we find seven distinct clusters. In the absence of genetic profiling, we cannot confirm the biological role of these clusters, but have demonstrated that morphology alone is enough to distinguish, cluster, and analyze cell sub-phenotypes.

phenotype ("normal" or invasive). We conclude that a simple morphodynamics test can reliably predict the results of a motility test, such as a *Boyden Chamber* test, i.e., segregate the motile from the non-motile cell populations. It appears, therefore, that using unsupervised clustering (and without any human input regarding classification into phenotypes), we can detect, with this morphodynamics based method, subpopulations of cells that are very different from the rest of the population, including their motility-related potential aggressiveness, i.e., *metastatic potential*. Thus, our technique augments, or obviates, the Boyden chamber assay, by allowing us to predict, using the morphodynamics analysis, which cells will migrate through the chamber. Additionally, we can detect sub-phenotypes of both migratory and non-migratory cells, though the origin of these sub-phenotypes remains uncharacterized. This preliminary evidence that new sub-phenotypes can be detected by our method will be a main item in our future studies.

## Discussion

Our *morphodynamic histology* approach pushes the biological adage, "structure dictates function". Using only a morphology analysis, we were able to cluster and differentiate (A) cells of separate and distinct lineages (e.g., MCF-7 vs. MDA-MB-231), (B) cells before and after having gone through the EMT (PC-3 vs. HR-14), and (C) cells from a single cell line (MDA-MB-231) that differed only in their motility. Our first classification experiment (Fig 2) is a proof of concept, as the ability to distinguish between the two genotypically distinct cell lines MCF-7 (epithelial-type) and MDA-MB-231 (mesenchymal-type) can be accomplished by eye. However, it is less trivial to delineate cells sharing a single genotype that have undergone the EMT. The second experiment (Fig 3) demonstrates that we can readily distinguish between pre- and post-EMT cells, suggesting that changes in phenotype are detectable as morphodynamic changes. In addition, the HR-14 cells have been shown, in animal models, to be much more metastatic than their epithelial counterparts [34]. In our final experiment (Fig 4), we used only a single population of cells, and we were able to identify, by morphodynamics alone, the highly motile cells within that population, as confirmed by a *Boyden* migration assay.

Importantly, we can track behaviors and morphological changes that are intrinsic to a *label free*, floating or circulating, cell, without relying on any biomarker or on any *a priori* knowledge of the genotype, which are often required for other CTC-capturing techniques [37, 38]. We note that *protrusions*, *blebs* and *amoeboid* morphologies have all been strongly correlated with an increase in malignancy [9, 34, 39, 40]. It thus follows that detecting such morphological cues could greatly help in the identification of those cells that are most responsible for the metastatic process. Importantly, early identification of cancerous or pre-cancerous morphodynamic phenotypes could enable early detection of cancers which produce no known biomarkers, such as is the case for pancreatic cancer [28, 41]. In future work, it would be beneficial to extend the dynamic morphodynamics platform to demonstrate sensitive differentiation of such cells, which could be of important diagnostic value.

Another advantage of the morphodynamic cell phenotyping approach is the ability to easily track thousands of individual cancer cells. Since the cells are loaded onto a grid of microfluidic wells, the location of each cell is known for the duration of the experiment, and the morphodynamic features of the cells can be tracked over time. While each present device is limited to a maximum capacity of 10,000 wells, the experimental approach is easily automatable, and there appears to be no difficulty in extending the analysis to other common model cell lines, or even to non-adherent cells.

To emphasize, in this study we have been able to start from a purely physical readout, i.e. the *shape features* of rotating cells, and end up segregating cells into morphodynamically unique clusters. While subtle differences in nanoparticle formulations have been shown to

effect cell responses to therapy [42, 43], our technique has been demonstrated to have minimal impact on cell viability (S2 Fig in S1 File). All along the analysis, our method maintains the ability to follow single cells individually. As such, this system lends itself well to serving as a single cell analysis assay. Importantly, each of the populations we studied (Figs 2–4) could be readily split into two distinct populations (e.g., MDA-MB-231 vs. MCF-7). Beyond that, our use of *unsupervised learning* demonstrated that there exist in these cell groups identifiable sub-populations of cells. The underlying genetic origin of these sub-clusters remains unresolved and their elucidation is beyond the scope of the present report. Indeed, their resolution may not be strictly fruitful as the inherent genetic instability of many late stage, metastatic cancers makes finding reliable, gene-based biomarkers quite challenging [44, 45]. Also, our technique, by probing morphology directly, avoids the difficulty of *attempting to associate unstable genetic codes with single, static cell states*.

Finally, we notice that by using just our morphodynamic analysis, we were able to easily distinguish the highly motile subpopulation of the MDA-MB-231 cells from that of the general population. As this entire cell line is considered to have a relatively high metastatic potential, and motility is often associated with intravasation, our analysis may have identified the more, or most, aggressive subpopulation of this cell line. Furthermore, we clearly demonstrate that morphology can be used to associate cells with real, physiological behaviors, such as crawling through the extracellular matrix (Boyden chamber). In our future work, our goal will be to establish the relationship between the biological heterogeneity and the morphological expression within a cell population, ultimately leading to the characterization of EMT activation within a cell population, without the need of any biomarker or genetic profiling.

In summary, we first introduced a new concept, *cell morphodynamics*, as well as the method for measuring it, based on the cell magneto-rotation (CMR) technique, which prevents cell adherence and allows 3-dimensional cell deformations, and on combining CMR with machine learning (ML) algorithms. This morphodynamic method is thus based on a *label-free testing* of non-fixed, minimally perturbed, live individual cells, kept in bio-mimetic micro-environments. Our massively parallel single cell analysis assay investigates the similarities and dissimilarities of cancer cells' morphological behaviors over time, and we could thus identify cells whose phenotype may be associated with a more, or most, malignant potential, including motility and invasiveness, and achieving this *without the use of any biomarker*. We further note that this approach does lend itself well to mapping the *heterogeneity characterizing a tumorous cell population*, as well as identifying the presence of both morphologically and biologically distinct subpopulations. We believe that these techniques could well present healthcare providers with a new and inexpensive tool for evaluating and predicting the *plasticity* and *potential aggressiveness* of a population or subpopulation of cancer cells, and how it might behave, without a genetic screen. We do plan, in future work, to further develop the method introduced here for the characterization of cellular subpopulations. This may benefit from sequencing specific single cells. Overall, this effort will be geared toward monitoring changes in the magnitudes and ratios of subpopulations of cell groups, so as to better predict a tumor's metastatic potential. We hope that such a rapid and reliable estimate of a tumor's migration potential could become an important feature of *informed precision cancer medicine*; it would provide the caregiver, as early as possible, with the likelihood of metastasis.

## Materials and methods

### Preparation of Magnetic Nanoparticles (MNPs)

Amine-coated magnetic nanoparticles (Ocean Nanotech®) with a diameter of 30nm, are prepared in a 1mL stock solution of 200μg/mL in cell culture media. We then add 15μL of Poly-

L-Lysine at 0.1%w/v (Sigma-Aldrich©), and the solution is left for an hour on a rotator at room temperature. The solution is then filtered using a 0.2$\mu$m syringe filter.

## Cell culture and magnetization

For these experiments, two lines of breast cancer cells, MCF-7 and MDA-MB-231, and one line of prostate cancer cells, PC-3, was used. These three cell lines were purchased from ATCC®. A fourth cell line, dubbed HR-14, which consisted of PC-3 cells that had undergone the EMT, was also used [34]. All cell lines were stably expressing Green Fluorescent Protein (GFP) and cultured in RPMI 1640 supplemented with 10% fetal bovine serum (FBS) and 1% Penicillin-Streptomycin-Glutamine (PSG) in a cell incubator at 37˚C, with 5% $CO_2$ and 100% humidity. Media and supplements were all purchased from Life Technologies©. Cells' confluency before addition of the MNPs is around 20–30%. Cells are incubated for 24 hours with cell culture medium to which is added (see below) 20$\mu$g/mL of amine-coated magnetic nanoparticles. These particles are uptaken via endocytosis (S1 Fig in S1 File).

## Microfluidic trapping system and cell loading

One hour before being exposed to fluorescent light, cells are washed with Hank's Balanced Salt Solution (HBSS) three times to remove traces of phenol red contained in the cell culture media, and then incubated for an hour in a colorless cell culture media that has been supplemented with the radical oxygen scavenger, Trolox (6-hyrdoxy-2,5,7,8-tetramethylchroman-2-carboxylic acid, Sigma-Aldrich), at 0.25nM. After an hour, cells are washed with HBSS, and gently detached using a cell scraper. Cell density is then adjusted by the help of a magnetic separator.

Cells are then gently pipetted into the microfluidic device. The microfluidic trapping device is made of polydymethylsiloxane, according to the protocol used by Park et al. [46]. Each well has a triangular shape, with a side size of 40$\mu$m and a depth of 35$\mu$m. The chip has two ports: An inlet port and an outlet port. Cells are loaded with a 100$\mu$L pipetter into the inlet, and gently introduced into the channel by applying negative pressure at the outlet. Once positioned, the device is put on top of a rare earth magnet to pull the cells down into the wells. We repeat these steps several times, until a sufficient loading ratio is achieved (above 60% of the traps occupied by single cells). These loading steps take around 3 minutes, and no more than 5 minutes. Finally, cells are washed with fresh media by gently pipetting fresh media into the device (fresh media is placed at the inlet port and pipetted from the outlet port). At the end of the imaging series (usually 60 minutes), propidium iodide is pipetted into the inlet port and the cells are imaged so that dead cells can be removed from the analysis.

## Cell Imaging and rotation

Cells are imaged on an Olympus©IX71™microscope, equipped with an arc-mercury lamp (U-RX-T™) and a high definition monochromatic digital camera (Q-Imaging©Retiga 6000, 10 Megapixels). To image simultaneously multiple positions of the device, the microscope stage is replaced with a motorized stage (ASI MS-4400 XYZ Automated Stage). Images are captured with the software package Micro-Manager (extension of ImageJ), while the stage is programmed and controlled via a custom made script in Micro-Manager [47]. To protect cells from light exposure, a custom made shutter opens for 700ms at every position each minute. Only single cells are kept to be measured. Temperature and humidity are controlled using a homemade, on-stage system that keeps the cells at 37˚C with 100% humidity. Cell media is supplemented with HEPES in order to maintain pH in the absence of $CO_2$. The oscillating magnetic field is generated via 4 solenoids positioned around and slightly above the

microfluidic device (S2 Fig in S1 File). All solenoids are driven by an alternating current with frequency of 15Hz; two solenoids are driven 90˚ out of phase. Suspended cells rotate with a frequency of 0.1Hz.

### Image processing

Raw images consist of a grid of cells at regular intervals (each cell is sitting in regularly-spaced microwells). Each live, single cell is cropped from the original image into separate, smaller images, each consisting of a single cell. It is these cropped images of single cells that are analyzed. The basis for measuring cell morphology relies on the accurate delineation of a cell's contours (S4 Fig in S1 File). This task is performed by a pipeline with the image analysis software CellProfiler [48]. Once cells are delineated, CellProfiler measures and records the value of different morphological parameters, such as cell area, perimeter, extent, etc., as well as Zernike moments and Haralick features. For a single experimental run, over 1000 individual cells are processed, and each cell has over 100 measured features.

### Data processing

To form a training set, the list of all the cells that have been monitored is established. Every time the classification function is called, the list of names (from different populations of cells that we want to distinguish) is re-shuffled and 70% of the cells are randomly selected to be part of the training set, the 30% left are kept as a testing set that is used to establish the efficiency of the algorithm; these percentages typically correspond to 1100 training cells and 400 tests cells. When a time limit is set, only the measurements at time points smaller than the time limit are kept to form the training and testing data sets. Once selected, the training set is normalized and its dimension reduced to 14 components with Principal Component Analysis (PCA). The parameters used for normalization and dimension reduction are kept and used to perform the same transformations on the testing set. To avoid over-fitting problems, we also used a Cross Validation technique, with a random shuffling of the data samples. A classifier is trained and results are calculated on the testing set. For a specific time limit and ratio between general and target population (i.e. MCF-7 vs MDA-MB-231), we repeat all these steps (from shuffling to testing) 30 times and average the results.

### Adaboost method—Supervised learning

In order to perform the machine learning step of our method, we split our data (morphology measurements) into two distinct subsets: a training subset and a testing subset. The training subset is used to train the computer to make decisions, while the testing set is used to evaluate its performance. We used 70% of the data as a training set, and the rest as a testing set, as is customary for machine learning problems [49, 50]. To avoid over-fitting, we also used a cross validation technique, with a random shuffling of the data samples. Cross validation trains the computer by using different training sets and evaluating its resolving abilities on the corresponding testing sets. This helps avoiding the problem of training the computer on a subset where the samples are too similar, leading to over-fitting, because the algorithm will be able to detect and rightly recognize only small variations from the training subset. In that case, when tested, the algorithm would perform poorly, and other variations would be missed. Shuffling cells randomly reduces the likelihood of this issue occurring. Finally, before we commence learning, the data is normalized and we perform a principal components analysis to reduce the dimensionality of the data from 169 to 14.

Training the computer means that for each entry, or cell measurement at a specific time point, we let it know the phenotype to which this entry belongs (0 if MCF-7, and 1 if

MDA-MB-231). Using the AdaBoost algorithm [33], the computer builds a decision procedure. When presented with unlabeled data (testing set), the decision rules are used to make predictions on the labels to assign whether a cell is an MCF-7 or an MDA-MB-231 cell. A way to measure the efficiency of a classification is by measuring its precision and recall, which are defined below:

$$Precision = \frac{T_P}{T_P + F_P} \tag{1}$$

$$Recall = \frac{T_P}{T_P + F_N} \tag{2}$$

Where the F1 score is the geometric mean of precision and recall:

$$F_1 score = 2\frac{Precision * Recall}{Precision + Recall} \tag{3}$$

Where $T_P$ and $F_P$ are true positives and false positives in the classification task, and $F_N$ represents the false negatives.

## K-means clustering—Unsupervised learning

Without indicating from which sample the cells came from (epithelial or mesenchymal), we used an unsupervised clustering algorithm to group similar cells together, and to find clusters. We then compared the clusters that were found with the actual sampled labels. To measure the accuracy of the fit, we used the homogeneity score. The homogeneity of the clustering measures whether each cluster contains only members of a single class (i.e. phenotype), and its value is between 0 and 1, where 1 means a perfect clustering and classification. Let A = $A_1$, $A_2$,...,$A_n$ be the true classes of data points that we have ("the ground truth"), and C = $C_1$, $C_2$,...,$C_l$ the classes obtained after clustering operations. We will set N to be the total number of data points. Let $a_m = \|A_m\|$ be the number of objects (i.e. cells) belonging to the m-th class, $c_k = \|C_k\|$ be the number of objects classified into the k-th cluster by the algorithm, and $n_{mk}$ the number of objects that belong to both $A_m$ and $C_k$. We can then define the homogeneity measure as:

$$homogeneity = 1 - E(A/C)/E(A) \tag{4}$$

where,

$$E(A/C) = -\sum_m \sum_k \frac{n_{mk}}{N} log \frac{n_{mk}}{a_k} \tag{5}$$

$$E(A) = -\sum_k \frac{a_k}{N} log \frac{a_k}{N} \tag{6}$$

Clustering was performed using the k-means algorithm [51]. The principle of this algorithm is to find clusters by minimizing the within-cluster sum of squares (WCSS). At first, k random points, called "means", are chosen, and for every single point left, the cluster to which it is attributed is the one where the WCSS is minimal. For each cluster formed, the means are calculated, and the attribution process is done again. These steps are repeated until the clusters are stable.

### Inducement of EMT in PC-3 cells

This epithelial to mesenchymal transition (EMT) protocol was originally developed by Roca et al [34]. Briefly, a subpopulation of PC-3 cells expressing luciferase and presenting an epithelial morphology were isolated. These cells were then co-cultured with interleukin-4 treated, CD14+ monocytes. These lines were cultured together for four days, which induced strong morphological changes in the PC-3 cell lines (PC-3-EMT). The PC-3-EMT cells were isolated and it was confirmed that concomitant with the morphological changes, the cells experienced a decline in E-cad expression while Vimentin expression had increased—changes consistent with cells having undergone the EMT.

### Cell migration assay

MDA-MB-231 cells were loaded into a standard Boyden chamber for cell migration assay [Cultrex Cell Migration Assay by R&D Systems]. After 12 hours, the highly motile cells that went completely through the porous membrane were detached and collected from the bottom part of the chamber. They were immediately loaded into the device and imaged with the help of fluorescence while being rotated.

## Supporting information

**S1 File. Methods and controls.** This document contains detailed description of the dynamic morphology method, both experimental and computational, as well as cell viability controls. (PDF)

## Acknowledgments

We thank Dr. Celina Kleer and Dr. Maria Elena Gonzalez of the University of Michigan's Department of Pathology for providing the GFP-expressing MCF-7 and MDA-MB-231 cells.

## Author Contributions

**Conceptualization:** Remy Elbez, Jeff Folz, Kenneth J. Pienta, Shuichi Takayama, Raoul Kopelman.

**Data curation:** Remy Elbez, Alan McLean.

**Formal analysis:** Remy Elbez, Alan McLean.

**Funding acquisition:** Shuichi Takayama, Raoul Kopelman.

**Investigation:** Remy Elbez, Hernan Roca, Shuichi Takayama, Raoul Kopelman.

**Methodology:** Remy Elbez, Shuichi Takayama, Raoul Kopelman.

**Project administration:** Remy Elbez, Jeff Folz, Shuichi Takayama, Raoul Kopelman.

**Resources:** Hernan Roca, Joseph M. Labuz, Kenneth J. Pienta, Shuichi Takayama, Raoul Kopelman.

**Software:** Remy Elbez.

**Supervision:** Shuichi Takayama, Raoul Kopelman.

**Visualization:** Joseph M. Labuz.

**Writing – original draft:** Remy Elbez, Jeff Folz.

**Writing – review & editing:** Jeff Folz, Alan McLean, Raoul Kopelman.

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
