## [Decision Letter · Decision Letter 0]

15 Sep 2021

PONE-D-21-19487Cell-morphodynamic phenotype classification with application to cancer metastasis using cell magnetorotation and machine-learningPLOS ONE

Dear Dr. Folz,

Thank you for submitting your manuscript to PLOS ONE. After careful consideration, we feel that it has merit but does not fully meet PLOS ONE’s publication criteria as it currently stands. Therefore, we invite you to submit a revised version of the manuscript that addresses the points raised during the review process. Please submit your revised manuscript by Oct 30 2021 11:59PM. If you will need more time than this to complete your revisions, please reply to this message or contact the journal office at plosone@plos.org. Please include the following items when submitting your revised manuscript:A rebuttal letter that responds to each point raised by the academic editor and reviewer(s). You should upload this letter as a separate file labeled 'Response to Reviewers'.A marked-up copy of your manuscript that highlights changes made to the original version. You should upload this as a separate file labeled 'Revised Manuscript with Track Changes'.An unmarked version of your revised paper without tracked changes. You should upload this as a separate file labeled 'Manuscript'.If applicable, we recommend that you deposit your laboratory protocols in protocols.io to enhance the reproducibility of your results. Protocols.io assigns your protocol its own identifier (DOI) so that it can be cited independently in the future. For instructions see: https://journals.plos.org/plosone/s/submission-guidelines#loc-laboratory-protocols. Additionally, PLOS ONE offers an option for publishing peer-reviewed Lab Protocol articles, which describe protocols hosted on protocols.io. Read more information on sharing protocols at https://plos.org/protocols?utm_medium=editorial-email&utm_source=authorletters&utm_campaign=protocols.

We look forward to receiving your revised manuscript.

Kind regards,

Sushanta K Banerjee, PhD

Academic Editor

PLOS ONE

4. Thank you for stating the following in the Acknowledgments/Funding Section of your manuscript:

“The authors wish to thank the National Institute of Health/National Cancer Institute 347 IMAT program for financial support through an NIH-NCI (IMAT) grant R21 CA160157 (RK), as 348 well as NIH grants CA136829 (ST), R01CA186769 (RK) and 1R01CA250499 (RK). JML gratefully 349 acknowledges support from the University of Michigan Tissue Engineering and Regenerative 350 Medicine Training Program (NIH T32-DE007057), a US Department of Education GAANN 351 fellowship, and the University of Michigan Microfluidics in Biomedical Sciences Training 352 Program (NIH T32 EB005582-05). The funders had no role in study design, data collection and 353 analysis, decision to publish, or preparation of the manuscript”

“The authors wish to thank the National Institute of Health/National Cancer Institute IMAT program for financial support through an NIH-NCI (IMAT) grant R21 CA160157 (RK), as well as NIH grants CA136829 (ST), R01CA186769 (RK) and 1R01CA250499 (RK). JML gratefully acknowledges support from the University of Michigan Tissue Engineering and Regenerative Medicine Training Program (NIH T32-DE007057), a US Department of Education GAANN fellowship, and the University of Michigan Microfluidics in Biomedical Sciences Training Program (NIH T32 EB005582-05). The funders had no role in study design, data collection and analysis, decision to publish, or preparation of the manuscript.”

Additional Editor Comments (if provided):

Reviewers' comments:

Reviewer's Responses to Questions

**Comments to the Author**

1. Is the manuscript technically sound, and do the data support the conclusions?

Reviewer #1: Yes

2. Has the statistical analysis been performed appropriately and rigorously? 

Reviewer #1: Yes

3. Have the authors made all data underlying the findings in their manuscript fully available?

Reviewer #1: Yes

4. Is the manuscript presented in an intelligible fashion and written in standard English?

Reviewer #1: Yes

5. Review Comments to the Author

Reviewer #1: The authors here report the use of microfluidics and machine learning to measure the shape and related dynamics and thus identify different phenotypes in non-adherent cells. They have used MDA-MB-231 and MCF-7 cells as well as those which have undergone EMT. They believe this method would provide rapid detection of metastatic cancer cells.

The authors have presented their data primarily on the two breast cancer cell lines with some insight into a prostrate cancer cell line. Can a healthy (non-cancerous) cell line be tested as a control to test their hypothesis?

Why were the magnetic nanoparticles used? Was it for growing spheroids? This portion must be clarified

Can the authors comment on what the significance of this method would be on cancers with a late prognosis with no early biomarkers available (pancreatic cancer)?

The following references should be added:

https://pubmed.ncbi.nlm.nih.gov/21553120/

https://pubmed.ncbi.nlm.nih.gov/33745223/

https://pubmed.ncbi.nlm.nih.gov/33151075/

https://pubs.acs.org/doi/abs/10.1021/jacs.6b12236

6. PLOS authors have the option to publish the peer review history of their article (what does this mean?). If published, this will include your full peer review and any attached files.

Reviewer #1: No

---

## [Author Response · Author response to Decision Letter 0]

14 Oct 2021

Response to Editor's Comment:

Our amended Funding Statement can be found below:

"The authors wish to thank the National Institute of Health/National Cancer Institute IMAT program for financial support through an NIH-NCI (IMAT) grant R21 CA160157 (RK), as well as NIH grants CA136829 (ST), R01CA186769 (RK) and 1R01CA250499 (RK). JML gratefully acknowledges support from the University of Michigan Tissue Engineering and Regenerative Medicine Training Program (NIH T32-DE007057), a US Department of Education GAANN fellowship, and the University of Michigan Microfluidics in Biomedical Sciences Training Program (NIH T32 EB005582-05). The funders had no role in study design, data collection and analysis, decision to publish, or preparation of the manuscript."

Response to Reviewers

1. The authors have presented their data primarily on the two breast cancer cell lines with some insight into a prostate cancer cell line. Can a healthy (non-cancerous) cell line be tested as a control to test their hypothesis?

We thank the reviewer for this comment, which is to verify that our technique can successfully resolve healthy and cancerous phenotypes. This task would necessitate the creation of a stable, GFP-expressing healthy breast line. Unfortunately, we have been unable to find healthy (non-cancerous) GFP-expressing epithelial or mesenchymal breast lines, such as within human mammary epithelial cells (HMECs), neither from cell vendors (e.g., ATTC) nor from laboratories within the University of Michigan. This would mean that we would need to create the GFP-expressing cell line within our laboratory from a non-GFP expressed base; unfortunately, while this process is possible through transfection, it will take an extended time, involve substantial costs, and would require control testing (serial passaging, etc.) once the GFP expression is achieved. We thank the reviewer for this insightful comment, and address it by having added several citations comparing cancer and non-cancerous cells. For example, Joshi et al. were able to differentiate triple negative breast cancer from healthy breast cells using an approach that combined machine learning with impedance microcytometry. Using impedance alone enabled successful classification of cancerous vs non-cancerous cells in 86.5% of instances. By including additional features such as phase change and current change, the success rate climbed to 97.3%. In another case, Hasan colleagues utilized time lapse fluorescent microscopy to track cell movements overtime. After extracting features form their images using a level-set algorithm, they were able to classify cancerous from non-cancerous cells using a naïve bayes classifier with 85% accuracy. Thus, we have added the following to the manuscript so as to highlight and make explicit these previous approaches, in contrast to ours:

“While previous studies have been conducted that differentiate cancerous from non-cancerous cells (27, 28), we have extended this kind of analysis so as to compare metastatic with non-metastatic cancer cell types.”

2. Why were the magnetic nanoparticles used? Was it for growing spheroids? This portion must be clarified

We thank the reviewer for this comment. To help clarify the role of the MNPs in this study, we have adjusted the introduction so as to include the following information:

“…, we use magnetic nanoparticles so as to trap, suspend, and rotate cells that are captured in a microfluidic chamber (29). Magnetorotation prevents cells from adhering the microwells and permits exploration of their morphodynamic space.”

Summarizing, in this study, the MNPs play no role in the formation of spheroids. They are used to prevent cells from adhering to the surface, so they can express their “shape-shifting” ability (see Fig S4 and previous group publication by Elbez. et. al. DOI: 10.1371/journal.pone.0028475 (ref 29)).

3. Can the authors comment on what the significance of this method would be on cancers with a late prognosis with no early biomarkers available (pancreatic cancer)?

We very much thank the reviewer for this comment. As our technique does not rely on specific biomarkers, but instead on the delineation of a cell’s morphology via fluorescence imaging, it may be able to identify pre-cancerous or cancerous morphological phenotypes present in pancreatic cancers. By detecting these phenotypes, our technique may be of tremendous diagnostic value by enabling their early detection. The following quote has thus been added to our discussion section:

“Importantly, early identification of cancerous or pre-cancerous morphodynamic phenotypes could enable early detection of cancers which produce no known biomarkers, such as is the case for pancreatic cancer (50, 51). In future work, it would be beneficial to extend the dynamic morphodynamics platform to demonstrate sensitive differentiation of such cells, which could be of important diagnostic value.”

4. The following references should be added:

https://pubmed.ncbi.nlm.nih.gov/21553120/

https://pubmed.ncbi.nlm.nih.gov/33745223/

https://pubmed.ncbi.nlm.nih.gov/33151075/

https://pubs.acs.org/doi/abs/10.1021/jacs.6b12236

We thank the reviewer for bringing these literature papers to our attention. These references have now all been added to the main text:

In the introduction:

“It has also been reported that morphological changes can be used to identify cells having undergone the EMT (10, https://pubmed.ncbi.nlm.nih.gov/21553120/).”

In the discussion:

“While subtle differences in nanoparticle formulations have been shown to effect cell responses to therapy (https://pubmed.ncbi.nlm.nih.gov/33745223/, https://pubmed.ncbi.nlm.nih.gov/33151075/), our technique has been demonstrated to have minimal impact on cell viability (Fig S2).”

“Importantly, we can track behaviors and morphological changes that are intrinsic to a label free, floating or circulating, cell, without relying on any biomarker, antibody, or on any a priori knowledge of the genotype, which are often required for other CTC-capturing techniques (https://pubs.acs.org/doi/abs/10.1021/jacs.6b12236, 38).”

In summary, we thank the reviewer for his many helpful comments, apologize for not being clear re the MNP role, and believe that we have now adjusted the manuscript so as to benefit from these comments.

---

## [Decision Letter · Decision Letter 1]

20 Oct 2021

Cell-morphodynamic phenotype classification with application to cancer metastasis using cell magnetorotation and machine-learning

PONE-D-21-19487R1

Dear Dr. Folz,

We’re pleased to inform you that your manuscript has been judged scientifically suitable for publication and will be formally accepted for publication once it meets all outstanding technical requirements.

Kind regards,

Sushanta K Banerjee, PhD

Academic Editor

PLOS ONE

Additional Editor Comments (optional):

Reviewers' comments:

Reviewer's Responses to Questions

**Comments to the Author**

1. If the authors have adequately addressed your comments raised in a previous round of review and you feel that this manuscript is now acceptable for publication, you may indicate that here to bypass the “Comments to the Author” section, enter your conflict of interest statement in the “Confidential to Editor” section, and submit your "Accept" recommendation.

Reviewer #1: All comments have been addressed

2. Is the manuscript technically sound, and do the data support the conclusions?

Reviewer #1: Yes

3. Has the statistical analysis been performed appropriately and rigorously? 

Reviewer #1: Yes

4. Have the authors made all data underlying the findings in their manuscript fully available?

Reviewer #1: Yes

5. Is the manuscript presented in an intelligible fashion and written in standard English?

Reviewer #1: Yes

6. Review Comments to the Author

Reviewer #1: All comments have been addressed by the authors.

They have done a good job in clearly explaining the role of the MNPs so as to avoid confusion.

The manuscript can be accepted in its current form

7. PLOS authors have the option to publish the peer review history of their article (what does this mean?). If published, this will include your full peer review and any attached files.

Reviewer #1: No

---

## [Editor Report · Acceptance letter]

4 Nov 2021

PONE-D-21-19487R1 

Cell-morphodynamic phenotype classification with application to cancer metastasis using cell magnetorotation and machine-learning 

Dear Dr. Kopelman:

I'm pleased to inform you that your manuscript has been deemed suitable for publication in PLOS ONE. Congratulations! Your manuscript is now with our production department. 

Kind regards, 

on behalf of

Professor Sushanta K Banerjee 

Academic Editor

PLOS ONE